# Ordered/Disordered Structures of Water at Solid/Liquid Interfaces

Chonghai Qi [1], Cheng Ling [2,3] and Chunlei Wang [4,*]

1 School of Physical Science and Intelligent Engineering, Jining University, Qufu 273155, China
2 Division of Interfacial Water and Key Laboratory of Interfacial Physics and Technology, Shanghai Institute of Applied Physics, Chinese Academy of Sciences, Shanghai 201800, China
3 University of Chinese Academy of Sciences, Beijing 100049, China
4 College of Sciences, Shanghai University, Shanghai 200444, China
* Correspondence: wangchunlei1982@shu.edu.cn

**Abstract:** Experiments and theory have revealed versatile possible phases for adsorbed and confined water on two-dimensional solid surfaces, which are closely related to the aspects of various phenomena in physics, chemistry, biology, and tribology. In this review, we summarize our recent works showing that the different water phases with disordered and ordered structures can greatly affect surface wetting behavior, dielectric properties, and frictions. This includes the ordered phase of water structure that induces an unexpected phenomenon, an "ordered water monolayer that does not completely wet water", at $T = 300$ K on the model's surface and some real, solid material, together with the anomalous low dielectric properties due to ordered water.

**Keywords:** ordered water monolayer; surface wetting behavior; dielectric properties; surface frictions

## 1. Introduction

An understanding of the interfacial water phase [1–9] is necessary for the understanding of various physical, chemical, and biological processes, such as the hydration of solutes in water [10–13], water adsorption behavior on solid surfaces [14–17], fluids and their transportation across channels [18–20], electrochemical reactions [21,22], protein stability and folding [23–27], and molecular peptide self-assembly [28,29]. Experiments and theories have suggested versatile possible phases for adsorbed and confined water [30–48], which are closely related to the aspects of various phenomena in the material sciences, such as geology, biology, tribology, and nanotechnology [10,49–51]. Unlike bulk liquids, water molecules strongly confined to or adsorbed on a solid surface usually exhibit a particular molecular structure and dynamic behavior. Due to the new high-resolution experimental technique developed in recent decades, the ordered phase of water molecules with a particular structure has been revealed [51–55]. In 1995, Hu et al. [54] firstly experimentally discovered the ordered water layer absorbed in the vicinity of the mica surface at room temperature. Subsequent work [56] by simulations and sum-frequency spectroscopy experiments performed by Miranda et al. [57] confirmed this two-dimensional hydrogen-bond network. Recently, atomic resolution-ordered water molecules have been directly observed on various materials' surface, thanks to high-resolution experimental techniques. Tetramer-ordered water molecules have been observed on the surface of NaCl(001) by Guo et al. [58] through the scanning tunneling microscope (STM) technique, at only 5 K. In 2019, Ma et al. found novel edge structures of 2D bilayer hexagonal ice appearing on the surface of Au(111) for the first time using noncontact atomic-force microscopy. Nanoconfinement, even in a hydrophobic space, can also induce ordered structures. Using high-resolution electron microscopy imaging, Geim et al. [59] experimentally observed the forms of square ice in between two graphene sheets at room temperature, a phase different from the conventional tetrahedral water structures forming about four hydrogen bonds (H-bonds). In theory,

several novel phase transitions of confined water were predicted. In 1997, Koga et al. [60] predicted a new bilayer ice phase form with interlocking H-bonds at extremely high pressures, and a new phase transition occurred when water molecules were confined between certain nanoscale spaces. In 2009, a new phase transition [61] between bilayer liquid and trilayer heterogeneous water confined in two parallel, atomically hydrophobic walls was observed, as water density increased. Interestingly, Han et al. [62] observed that the first-order transition line of water molecules confined between hydrophobic space can connect with a continuous transition line, instead of terminating at a critical point. Kapil et al. [63] combined state-of-the-art computational approaches to perform a first-principles-level study on a water monolayer confined within a graphene-like channel. They observed that this water layer exhibited interestingly abundant and various phase behaviors, which was greatly dependent on the pressure and temperature. These results suggest that confined water exhibits versatile and complex phase diagrams. In respect of the biology, interfacial phase water that is strongly associated with biomolecules has been observed near the biological molecules [64–68], which may play key roles in the function and diffusion of the biomolecules.

In the present review, we summarize the recent works showing that the phase transition of water from disordered to ordered structures in the first layer to contact with surface can greatly affect surface wetting behavior, surface frictions, and dielectric properties. We highlight the distinct phases of water structures' contact with the surfaces with different H-bond networks, which are mainly attributed to the matching or mismatching between the water structure and the atomic arrangement of a solid surface. The article is organized in four sections. In Section 2, we show the prediction that the hydrophobicity of an ordered water monolayer can be unexpectedly enhanced by ordered water structure at room temperature, which is referred to as an "ordered water monolayer that does not completely wet water". In Section 3, we show the novel wetting behavior of water on the COOH matrix and SAM-$(OH)_2$. In Section 4, we show the dependence of the ordered water structure on parallel dielectric permittivity. In the last section, we provide a summary and the future directions of the research.

## 2. Phase Transition from Disordered to Ordered Water Structures Induces an "Ordered Water Monolayer That Does Not Completely Wet Water" at Room Temperature on Solid Surfaces

The hydrophobic-like ice monolayer at cryogenic temperatures [69] was not expected at room temperature, while room temperature, strong, thermal fluctuations usually disturb H-bond networks in water. It is not difficult to rationalize that water at room temperature plays a greater role in daily life and applications, including surface friction on solid/water interfaces, water warming or cooling, solvation of solvent molecules, biological activities and functions, and oil mining underground. Since hydrogen bonds form among water molecules, water molecules are naturally and always completely wetting other water. Figure 1a presents our molecular dynamics (MD) prediction that a water droplet with a pronounced contact angle stands on a very thin (0.4 nm) water monolayer in the vicinity of an ionic solid surface at room temperature. This phenomenon has been referred to as an "ordered water monolayer that does not completely wet water" at room temperature [70–75]. In Figure 1b, we present a theoretical ionic model surface with a planar hexagonal charge pattern structure with charge values $q$ at certain locations, while the surface is neutral in total.

This phenomenon is mainly attributed to the ordered phase water structures resulting from the high charge and special arrangement of the solid surface, which decrease the number of H-bonds in the inter-layer and increase the number of H-bonds in the intra-layer. Our calculation shows that the thermal conductivity of the ordered water monolayer would most likely resemble ice rather than liquid water, from a thermodynamics viewpoint [76]. The solid lattice structure, such as the surface bond length $l$, is fundamental to the phase transition of disordered to ordered water structures [70,74], which can transform the nonwetted to a completely wetted water layer. $l$ increases from 0.142 to 0.16 nm or decreases

from 0.142 to 0.12 nm; we observed there were no ordered water structures, whereas charge $q$ was high enough (1.0 e). Our recent study showed that the ordered–disordered phase transition of interfacial water was considerably relevant to the charge dipole moment, production of both charge values, and the dipole length of the solid surface. In addition, the surface point defects [74], temperature [77], and curvature [78] can also induce the wetting transition from nonwetted to completely wetted due to the disruption of ordered water structures. In 2015, we observed a 25% friction reduction on the super-hydrophilic materials due to the ordered water molecular structures [79]. This further demonstrates the important role of ordered water structures in hydrodynamics. Recently, the water-induced friction reduction was experimentally verified by Ma et al. on $TiO_2$ silica surfaces [80] and other solid surfaces [81]. This super-hydrophilic but low-friction surface may have great application potential in self-clearing materials [82–85], biomedical materials [86,87], and nano-coating materials of ships [82].

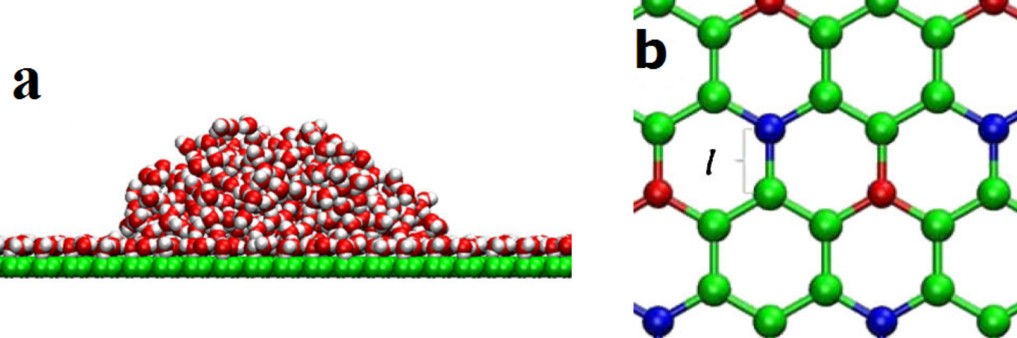

**Figure 1.** (**a**) A stable water droplet coexists with a thin water monolayer absorbed on a modeled solid surface. (**b**) Structure of the model solid surface with a hexagonal, charged pattern. Red presents the positive-charged atoms, and the blue presents the negative-charged atoms. (Reprinted from Ref. [70], Copyright 2009 American Physical Society).

Recently, we applied this therical framework to solve the long-debated question of understanding wetting behaviors on $TiO_2$ surfaces [88]. Since the first experimental observation of the wetting transformation on a solid surface under UV exposure in 1997, the wetting of a $TiO_2$ surface has been debated for decades [89,90]. We noted that the macroscopic contact angle was around 15–30° initially on a freshly prepared $TiO_2$ surface, but increased to 60–70° under ambient conditions in the dark [83,89–94]. However, water droplets diffuse all over the film following UV exposure, leading to super-hydrophilic behavior with a contact angle of 0°. Then, several works determined that the existence of an intrinsically hydrophobic (oleophilic) region [89,90] or adsorbed hydrophobic hydro-carbon contamination [95,96] contributed to nonwetting behavior. In 2018, Diebold et al. claimed that the amphiphilic carboxylic acid monolayer on rutile $TiO_2$(110)'s surface could induce hydrophobic behavior through the experimental method combining atomic-scale microscopy (AFM), a scanning tunneling microscope (STM), and X-ray photoelectron spectroscopy (XPS) [97]. In fact, molecular-ordered water structures were observed in some theoretical works by MD simulations [91,98] and experiments under vacuum conditions or at cryogenic temperatures [99–105]. However, the relationship between water's structure and wetting behavior was not carefully considered. Intuitively, the interactions (>1.0 eV) between $TiO_2$ and water [106] are strong; thus, in principle, a super-hydrophilic surface similar to a mica surface [54,56] is expected. We then speculated that the unexpected wetting with a large contact angle can be exactly solved using our previous model shown in Figure 1. We used MD simulations with a classical force field and neural network potential (NN-MD) to firstly identify a water droplet on an ordered water bilayer structure in the vicinity of a rutile $TiO_2$ surface under ambient conditions (see Figure 2a,b). The ordered water structure decreased the number of H-bonds between the bilayer and water droplet,

and thus created an obvious contact angle. UV exposure with a reduction in the quantity of adsorbed molecular water and an enhancement in the quantity of adsorbed dissociated water with −OH at the surface was evident. Therefore, we investigated the effect of 5% and 10% covering ratios of −OH groups on the wetting behavior of rutile $TiO_2$(110)'s surface, with the contact angle of the water droplet on the water bilayer decreasing from 19° (5%) to 0° (10%). This can be ascribed to the disruption of the water bilayer H-bonds network, which transforms the hydrophobicity of the water bilayer to super-hydrophilic.

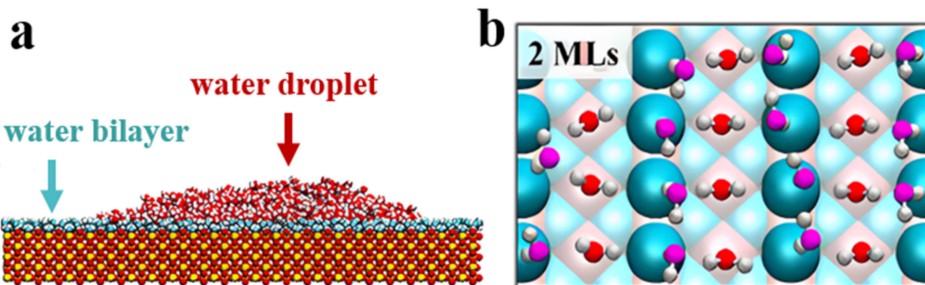

**Figure 2.** (**a**) Side-view snapshot of the rutile $TiO_2$(110) solid with a water droplet (red and white balls) coexisting with the water bilayer (cyan and white balls). (**b**) Snapshots of the ordered water structures for 2 MLs of $H_2O$ adsorbed on rutile $TiO_2$ (110). The atoms are color coded as follows: Ti (pink), O of $TiO_2$ (cyan), O of $H_2O$ (first layer: red; second layer: magenta), and H (white). (Reprinted from Ref. [88], Copyright 2022 Royal Society of Chemistry).

Since 2009, many research groups have provided proof of this similar phenomenon [107], suggesting it extensively exists on several real materials' surfaces. The majority of these works utilized the physical mechanism of the phase transition of water from disordered to ordered structures; we proposed to explain their observations in theoretical simulations or experiments. In 2015, we observed that this phenomenon may exist on Pd(100) (with a contact angle value of 57°), Pt(100) (with a contact angle value of 53°), and Al(100) (with a contact angle value of 32°) surfaces, while it cannot be observed on other solid-metal surfaces, such as (110) and (111) surfaces (see Figure 3c,e,f) and the (100) surface of Ni with a large lattice constant [108]. We observed that the clear, ordered water structure (see Figure 3g) induced the wetting transition from a completely wetted to a nonwetted water monolayer. As shown in Figure 3h, for various surfaces with different metal surfaces with different lattice constants, four clear orientation preferences of the water dipole are observed for Pd(100) and Pt(100) surfaces at $\varphi = 0°$, 90°, 180°, and $\varphi = 270°$, respectively. This indicates that the formation of ordered water structures is induced by the matching between the lattice structure for Pd(100) and Pt(100) surfaces and water H-bonds. Interestingly, these ordered water structures show rhombic structures, different from the previous hexagonal structures [70].

In 2011, Rotenberg et al. [109] showed the wetting behavior of a talc surface dependent on humidity using MD simulations, i.e., hydrophilic at low humidity levels, while hydrophobic at high humidity levels. This large contact angle is consistent with experiments that show that the macroscopic contact angle is about 80°–85° [110]. Then, Phan et al. [111] observed the phenomenon of an "ordered water monolayer that does not completely wet water" on hydroxylated $SiO_2$ (111) and $Al_2O_3$ surfaces utilizing MD simulations. They directly used the physical mechanism of the disordered–ordered phase transition of interfacial water structures to explain the observations. In 2021, we observed the wetting phenomenon with a droplet on composite structures formed by embedded water in the (111) surface of $\beta$-cristobalite hydroxylated silica [112]. In 2013, Limmer et al. [21] also observed the novel wetting phenomenon that an "ordered water monolayer that does not completely wet water" on the Pt(100) surface. Note that each water sample in the ordered monolayer tends to form about four H-bonds, which is different from the hexagonal structure of the ordered water monolayer with three H-bonds on our model surface [70,71,113].

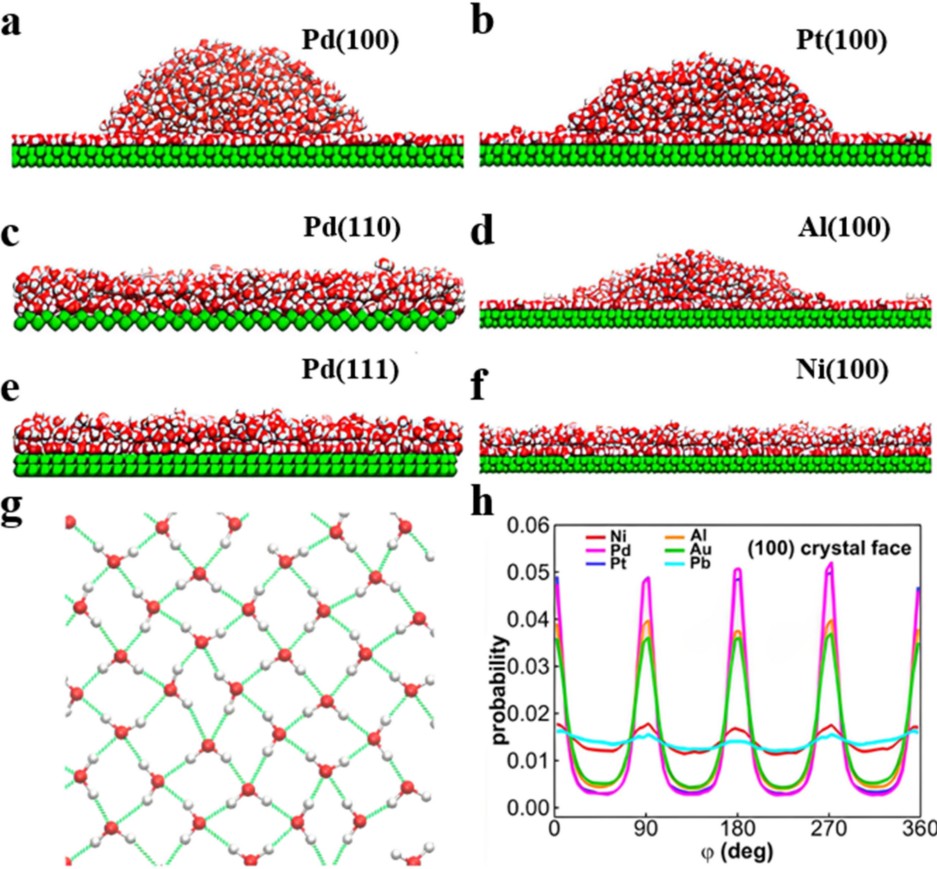

**Figure 3.** Typical side view of a water droplet on an ordered water monolayer on (**a**) Pd(100), (**b**) Pt(100), and (**d**) Al(100) surfaces, and the spreading water film on some typical surfaces: (**c**) Pd(110), (**e**) Pd(111), and (**f**) Ni(100). (**g**) Snapshot of the ordered, rhombic water molecules (in green lines) forming between neighboring water molecules. (**h**) Probability distribution of angle $\varphi$ between the x–y plane projection of one water molecule dipole orientation and x-axis for (100) the crystal face. (Reprinted from Ref. [107], Copyright 2015 American Chemical Society).

In addition to the simulations, in respect of the experiments, Lützenkirchen et al. [114,115] observed an ice-like configuration of water molecules on the sapphire *c*-plane surface by using the sum frequency generation (SFG) technique, which was distinct from the bulk liquid water phase. It should be noted that the contact angles of the liquid water droplets on the solid surface were quite large [114]. In 2014, Lee et al. [98] used the mechanism inspired by our study to explain experimental superwetting under light-illumination conditions on Titania surfaces. In 2021, direct experimental evidence for the observation of this ordered water layer on surfaces, particularly on biomolecule and polymer surfaces, was validated at room temperature for a hydrophobic fluorinated polymer, such as polytetrafluoroethylene's (PTFE's) surface [116], by employing SFG vibrational spectroscopy. These experimental evidence provides the platform for future applications in terms of the related materials.

### 3. Ordered Phase of Composite Structures of Water Molecules Embedded into the Carboxylic Acid-Terminated Self-Assembled Monolayers (COOH-SAMs) and Hydroxyl-Terminated Self-Monolayer ((OH)$_2$-SAMs) at Room Temperature

Carboxylic acid-terminated self-assembled monolayers have attracted considerable interest due to their wide applications in nanoscience and nanotechnology [117–125]. However, there were inconsistent values of water contact angles on COOH-SAMs, even after 25 years of study, which still puzzles researches regarding the surface water adsorption behavior on COOH-SAMs. We collected these contact angle values and observed that

the values fell in the range of 0° to 50° from forty literature experiments [118,126–129]. Particularly, in 2011, James et al. [130] observed water droplets coexisting with a continuous few-angstrom-thin water layer on COOH-SAM using X-ray, neutron reflectometry, and atomic force microscopy (AFM) methods. Moreover, the adsorbed water molecules at the COOH-SAM surface showed novel adsorption behaviors, different from the conventional viewpoint. A similar situation can be observed for SAM-OH surfaces, which are usually regarded as super-hydrophilic.

We observed that water molecules can be embedded in the COOH matrix on COOH-SAMs with appropriate packing densities to form an ordered phase structure of an embedded water–COOH composite [131], which enhances the hydrophobicity of embedded water–COOH composite structures (see Figure 4a). Remarkably, a liquid water droplet with an apparent contact angle of approximately 34° stands on the embedded water–COOH composite [Figure 4a]. This phenomenon is caused by the water embedded in COOH groups forming an H-bond network, which leads to reduced H-bond numbers between the surface and water molecules above the composite structure. This can explain the experimental work by James et al. well. Figure 4b shows the contact angles of water droplets on COOH-SAMs as a function of density $\Sigma$. There is an angle plateau of ~35° in the mid-range and a value ~0° in the dense range. In the sparse range, we observed the contact angle $\theta$ initially decreased and then slightly increased as $\Sigma$ increased. In 2013, Wang et al. [132] experimentally observed the phenomenon of water droplets coexisting with a nanoscale water layer formed on bovine serum albumin (BSA) when the IB was sealed at a low RH (15–25%) at room temperature for 3–5 days.

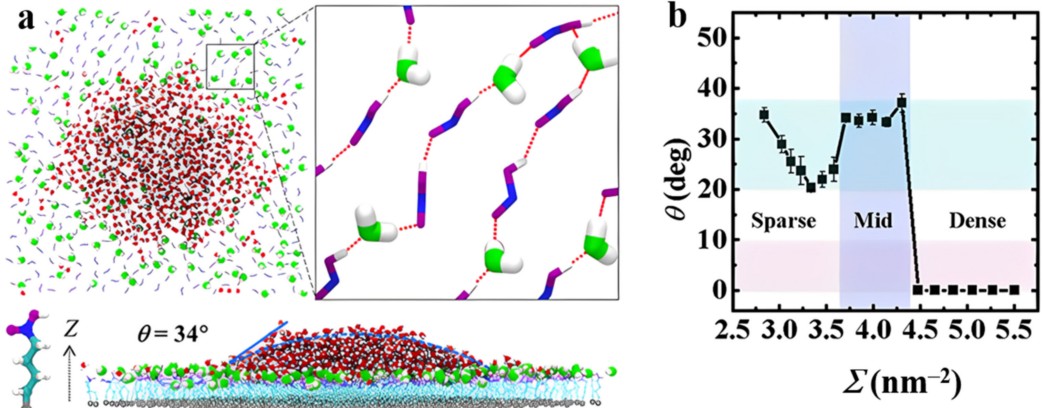

**Figure 4.** (**a**) Side-view snapshots of water droplets and sparse water molecules outside the droplet on COOH-SAM with a packing density of $\Sigma$ = 4.00 nm$^{-2}$, together with top and side view snapshots of water on COOH SAM (model atoms: gray; COOH groups: blue, purple, and white; water: red and white; embedded water: green and white; alkyl chains: blue lines in side view, but omitted for clear views in the top view; H-bonds: red, dashed lines). (**b**) Dependence of contact angle values $\theta$ of water droplets on packing density $\Sigma$ of COOH-SAMs. We marked the mid-range in light blue. (Reprinted from Ref. [131], Copyright 2015 American Physical Society).

In addition to the hydrophilic group COOH, there are very few reports on how to obtain hydrophobicity on surfaces constituted by only utilizing hydrophilic groups/molecules. Our recent study determined that a typical hydrophilic -OH group can also induce a hydrophobic surface with large contact angles. In our study. we used five-carbon long alkyl chains, which were grafted on one end with two -OH groups exposed to water. The packing density ($\Sigma$) varied from 2.0 to 6.5 nm$^{-2}$. Then, we performed MD simulations [133] to analyze the water droplet wetting behavior and calculated the contact angles of water droplets on (OH)$_2$-SAMs. Figure 5a presents a representative snapshot of a water droplet on (OH)$_2$-SAM for $\Sigma$ = 4.5 nm$^{-2}$, where the contact angle of the water droplet is 82°, indicating the hydrophobicity of (OH)$_2$-SAM. However, the average number of H-bonds in

between the neighboring OH groups is approximately 8.8 nm$^{-2}$, suggesting that even the typical hydrophilic OH group can exhibit hydrophobic behavior. We also show that the contact angles of water droplets on (OH)$_2$-SAMs depend on the function of packing density $\Sigma$, similar to COOH-SAM. Interestingly, the water molecules can be embedded into the OH groups to form composite structures with looser packing densities $\Sigma$. This composite structures also enhanced the hydrophobicity of (OH)$_2$-SAM.

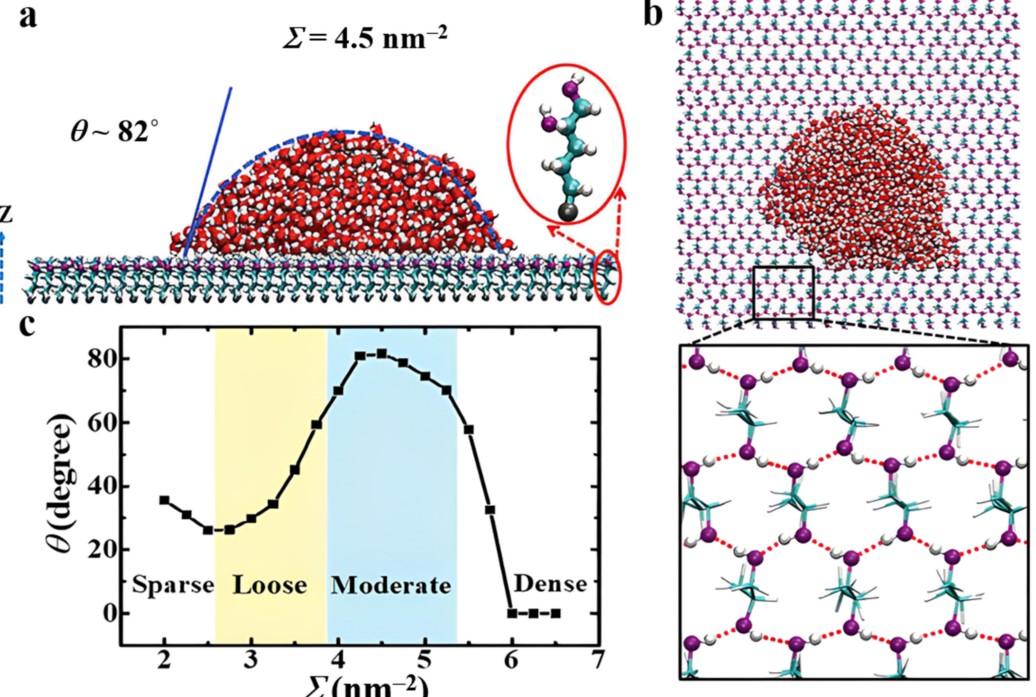

**Figure 5.** (**a**) Side-view snapshot of a water droplet with large contact angle of 82° on (OH)$_2$-SAM with two hydroxyl (OH) groups at a packing density of $\Sigma$ = 4.5 nm$^{-2}$ (model atoms, gray; hydroxyl groups: purple and white; water: red and white; and alkyl chains: cyan and white). (**b**) Top-view snapshot (top) of the subfigure (bottom) with an enlarged region where a hexagonal-like H-bonding structure on (OH)$_2$-SAM appears. (**c**) Contact angle $\theta$ of the water droplets on (OH)$_2$-SAMs versus packing density $\Sigma$. (Reprinted from Ref. [133], Copyright 2021 Royal Society of Chemistry).

## 4. Effect of Ordered Phase of Water on the Dielectric Permittivity

Compared with other fluids, bulk water has a large, static, dielectric constant, which is very important in various aqueous environments, such as energy systems' [134,135] biomolecule function [136,137] and ion-ion interactions [55,138,139]. In confinement or close to the interface, the dielectric properties of water become anisotropic [136,140–146]. For many decades, water's dielectric permittivity has been intensively studied [136,140–145,147–153]. Recently, water's dielectric permittivity in a perpendicular direction between hBN and graphene nanochannels has been measured [141]. For the lower values of interslab separation, an anomalously low perpendicular dielectric constant (as low as 2) was reported. In addition to nanoconfinement, the number density of water molecules [145,146] or surface wettability [140,154] were also reported to impact the dielectric permittivity of water close to solid surfaces.

Recent works have shown that an ordered water structure can form in the vicinity of some solid surfaces at room temperature, such as ionic model surfaces, metal surfaces [21,78,155,156], metal oxides [117,157], and clay surfaces [109]. Our MD simulations revealed that the parallel dielectric permittivity of interfacial water depends on solid-water interactions together with the interfacial water structure on various sold crystal

faces [158]. In particular, the in-plane dielectric permittivity of ordered water structures on solid surfaces can be significantly reduced.

The different trends for the parallel dielectric constant of water ($\varepsilon_\parallel^{\text{Interfacial}}$) close to (100) and (111) surfaces versus the different surface-water interactions ($f$) are shown in Figure 6. For the fcc (100) surface, $\varepsilon_\parallel^{\text{Interfacial}}$ initially increases from 60 to 102 and then reduces to 40. This sudden decrease in $\varepsilon_\parallel^{\text{Interfacial}}$ originates from the ordered H-bond network (see Figure 3g), leading to the low amplitude of the dipolar fluctuation. This nonmonotonic behavior of $\varepsilon_\parallel^{\text{Interfacial}}$ with $f$ is quite different from previous reports that show that the more hydrophilicity on the surface, the larger the dielectric constant [145,146]. However, for the fcc (111) surface, $\varepsilon_\parallel^{\text{Interfacial}}$ increases from 78 to 129 with $f$, in accordance with the previous reports [145,146]. Our work indicates that the crystal orientation of hydrophilic surfaces can significantly affect the parallel dielectric permittivity of interfacial water.

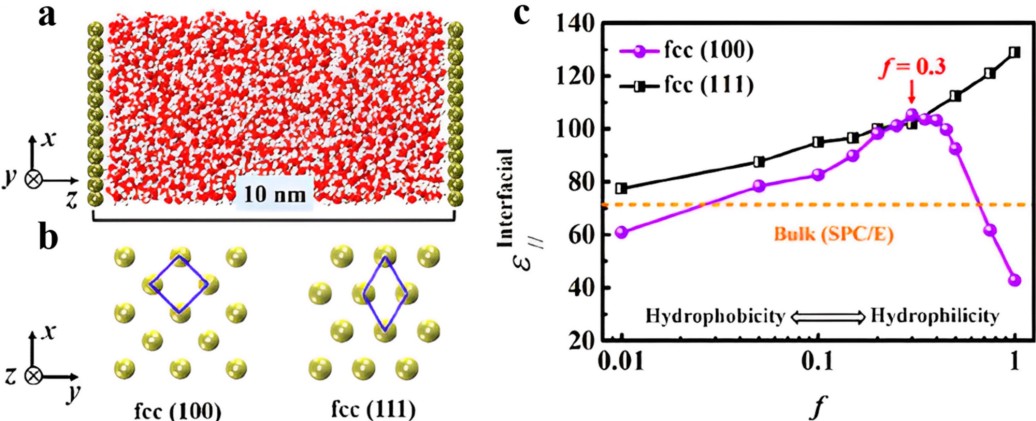

**Figure 6.** Anomalously low, parallel, dielectric constant of interfacial water at the fcc (100) hydrophilic surface: (**a**) schematic representation of the confined water between two fcc (100) or (111) sheets separated by a distance of 10.0 nm in the z direction. (**b**) Top view of the lattice arrangement of the fcc (100) (left) and (111) (right) surface models, and (**c**) parallel permittivity of the dielectric constant of interfacial water $\varepsilon_\parallel^{\text{Interfacial}}$ (equal to $\varepsilon_{xx}$ and $\varepsilon_{yy}$) at the fcc (100) and (111) surfaces versus the surface-water interaction $f$ (the orange, dashed line shows the constant for bulk water). (Reprinted from Ref. [158], Copyright 2021 American Chemical Society).

## 5. Summary

In the current review, we summarized the recent advances in how the phase transition of water from disordered to ordered structures in the first layer to contact with the solid surface affects a variety of surface properties, such as wetting behaviors, surface dielectric properties, and surface frictions. The ordered phases of water structure induce unexpected phenomenon, an "ordered water monolayer that does not completely wet water" at room temperature on ionic model solid surfaces, and metals, metal oxides, and minerals. We used this theorical framework to solve the long-debated question of understanding wetting behaviors on TiO$_2$ surfaces. The ordered phase of composite structures of water molecules embedded in the carboxylic acid-terminated self-assembled monolayers (COOH-SAMs) enhances hydrophobicity at room temperature. Similarly, the hydrogen-bond network of hydroxyl-terminated self-assembled monolayers ((OH)$_2$-SAMs) and water molecules embedded in (OH)$_2$-SAMs also induce unexpected hydrophobicity at room temperatures. Particularly, those wetting behaviors previously described exhibit clear water droplets at a macroscopic level indicating hydrophobicity, but the water layer at a molecular level indicates hydrophilicity, which is termed "molecular-scale hydrophilicity" [159]. This transition from a disordered to ordered water structure is mainly attributed to the mismatching or matching between solid surface structures and water molecules' surfaces. Our works show that the various microscopic water structures with variable quantity allocations of H-bond

numbers between water molecules can regulate physical interactions, even macroscopic properties, i.e., wetting, friction reduction, and dielectric properties.

Despite the considerable efforts made, there are still many unanswered questions in this research field. Recent simulations and experimental works [88,98,114–116,130,132] have determined the phenomenon that an "ordered water monolayer that does not completely wet water"; however, whether there will be more materials found in this direction is still unknown. We noted that more direct results for ordered water structures that can lead to wetting behavior are still lacking at room temperature, which may be attributed to the lack of accuracy of the resolution in techniques at present. Subsequently, how these ordered water structures on solid metal or metal oxide surfaces affect surface properties, such as the catalysis, non-fouling, electric potential, and gas formation, is still in urgent need of research. For example, whether the ordered water hydrogen-bond network affects water transport through covalent organic framework channels [18,160], the evaporation of interfacial water [161] and lubrication strategy of hydrogels [162] is still an open question in terms of the applications. The so-called physical electric double layer (EDL) was usually induced by the charged surfaces when an aqueous electrolyte solution was present. However, the classical theory neglecting the interfacial molecular water structures is still unable to fully describe the observed phenomenon under the assumptions, such as no ion–ion correlations and the homogeneous dielectric continuum of water [163]. It should be noted that the quantum nuclei effect should be given more attention when introducing important effects on the hydrogen bonds and reorientation of water [164,165] spectra in terms of both the energy and line shapes [166].

**Author Contributions:** Conceptualization, C.Q. and C.W.; Methodology, C.W.; Software, C.Q. and C.L.; Validation, C.Q. and C.L.; formal analysis, C.Q.; investigation, C.Q. and C.W.; resources, C.W.; data curation, C.Q. and C.W.; writing—original draft preparation, C.Q. and C.W.; writing—review and editing, C.Q. and C.W.; supervision, C.W.; funding acquisition, C.Q. and C.W. All authors have read and agreed to the published version of the manuscript.

**Funding:** This work was supported by the National Science Foundation of China (Grant Nos. 12022508, 12074394, 11674345), the Key Research Program of the Chinese Academy of Sciences (QYZDJ-SSW-SLH019) and the Natural Science Foundation of Shandong Province of China (Grant No. ZR2022QA089).

**Data Availability Statement:** The data presented in this study are available on request.

**Acknowledgments:** Computations were performed on the Shanghai Supercomputer Center of China.

**Conflicts of Interest:** The authors declare no conflict of interest.

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
