# Peer review of "Ordered/Disordered Structures of Water at Solid/Liquid Interfaces"

_crystals, doi:10.3390/cryst13020263_

Round 1

Reviewer 1 Report

This is a really well written work that explains how experiments and theory have shown that adsorbed and condensed water at two-dimensional solid surfaces can adopt a range of phases that are closely related to a number of phenomena in physics, chemistry, biology, and tribology. Surface wetting behavior, surface dielectric properties, and surface frictions can all be greatly influenced by the many water phases, which can have both disordered and ordered structures. 

My biggest complaint is that the body of work that is cited does not acknowledge earlier work and gives priority to the work of the authors. The groundbreaking research of Professor Jacob N. Israelachvili, who wrote the book "Intermolecular and Surface Forces," provides proof of the phenomena the writers describe. The development of experiments and theory has advanced, however attributing the inception of the investigation to the writers is misleading. I have published a lot of work myself creating AFM-based tools to look at the function of confined water. Here, it is a proof of few of these overlooked works by the authors:

A nanoscopic approach to studying evolution in graphene wettability CY Lai, TC Tang, CA Amadei, AJ Marsden, A Verdaguer, N Wilson, ...Carbon 80, 784-792 77 2014

Time dependent wettability of graphite upon ambient exposure: The role of water adsorption CA Amadei, CY Lai, D Heskes, M Chiesa The Journal of chemical physics 141 (8), 084709

The aging of a surface and the evolution of conservative and dissipative nanoscale interactions CA Amadei, TC Tang, M Chiesa, S Santos The Journal of chemical physics 139 (8), 084708

The evolution in graphitic surface wettability with first-principles quantum simulations: the counterintuitive role of water JY Lu, CY Lai, I Almansoori, M Chiesa Physical Chemistry Chemical Physics 20 (35), 22636-22644

Author Response

This is a really well written work that explains how experiments and theory have shown that adsorbed and condensed water at two-dimensional solid surfaces can adopt a range of phases that are closely related to a number of phenomena in physics, chemistry, biology, and tribology. Surface wetting behavior, surface dielectric properties, and surface frictions can all be greatly influenced by the many water phases, which can have both disordered and ordered structures. 

Author Reply:

We thank the Reviewer for his/her positive comments and constructive suggestions.  We have also carefully revised our manuscript accordingly. Our detailed responses to the questions are as follows.

My biggest complaint is that the body of work that is cited does not acknowledge earlier work and gives priority to the work of the authors. The groundbreaking research of Professor Jacob N. Israelachvili, who wrote the book "Intermolecular and Surface Forces," provides proof of the phenomena the writers describe. The development of experiments and theory has advanced, however attributing the inception of the investigation to the writers is misleading. I have published a lot of work myself creating AFM-based tools to look at the function of confined water. Here, it is a proof of few of these overlooked works by the authors:

A nanoscopic approach to studying evolution in graphene wettability CY Lai, TC Tang, CA Amadei, AJ Marsden, A Verdaguer, N Wilson, ...Carbon 80, 784-792 77 2014

Time dependent wettability of graphite upon ambient exposure: The role of water adsorption CA Amadei, CY Lai, D Heskes, M Chiesa The Journal of chemical physics 141 (8), 084709

The aging of a surface and the evolution of conservative and dissipative nanoscale interactions CA Amadei, TC Tang, M Chiesa, S Santos The Journal of chemical physics 139 (8), 084708

The evolution in graphitic surface wettability with first-principles quantum simulations: the counterintuitive role of water JY Lu, CY Lai, I Almansoori, M Chiesa Physical Chemistry Chemical Physics 20 (35), 22636-22644

Response: Thank you for your valuable comments. We have revised our manuscript by adding “Since 2009, many research groups have provide the proofs of this similar phenomenon, suggesting the similar phenomenon extensively exist on several real materials surfaces.” on line 20 of page 4, and “Experiments and theory have suggested versatile possible phases for adsorbed and confined water [30–48] which are closely to the aspects of various phenomena in materials science, geology, biology, tribology and nanotechnology.[10,49–51]” on line 27 of page 1.

Reviewer 2 Report

In this manuscript, the authors review a large number of molecular simulation works for water structures and behaviors on solid surfaces, that were originally done by the authors and also many other researchers. The review is extensive and well-written. I really enjoyed reading every section of the manuscript. A few suggestions to improve the manuscript:

1. Although the review mainly discusses realistic models of liquid/solid systems, it is always interesting to touch some discussion on simple yet fundamental issues such as the dependency of the contact angle on water models (doi:10.34311/icspc.2021.1.1.10), the method used for calculating contact angle of a water droplet (doi:10.1063/1.4978497), wetting on smooth versus patterned solid (doi:10.1021/acs.jpcc.8b01344).

2. In Figure 6, the authors discuss an interesting relationship between the water dipole moment with the change in the structure of the surface where the water is attached. It will be helpful if the author can address whether the intermolecular distance between surface atoms (i.e. the solid density) also affects the dipole moment.

3. Not a critical issue, but I hope that the publisher, in particular the layout team, can help to typeset the "\varepsilon ||^{interfacial}" symbol because, in its current state, the symbol does not look well in Figure 6 caption (and on paragraph too). Alternatively, the authors can choose to simplify the symbol.

Looking forward to seeing the review published.

Author Response

In this manuscript, the authors review a large number of molecular simulation works for water structures and behaviors on solid surfaces, that were originally done by the authors and also many other researchers. The review is extensive and well-written. I really enjoyed reading every section of the manuscript. A few suggestions to improve the manuscript:

Author Reply:

We thank the Reviewer for his/her positive comments and constructive suggestions.  We have also carefully revised our manuscript accordingly. Our detailed responses to the questions are as follows.

Point 1. Although the review mainly discusses realistic models of liquid/solid systems, it is always interesting to touch some discussion on simple yet fundamental issues such as the dependency of the contact angle on water models (doi:10.34311/icspc.2021.1.1.10), the method used for calculating contact angle of a water droplet (doi:10.1063/1.4978497), wetting on smooth versus patterned solid (doi:10.1021/acs.jpcc.8b01344).

Response1: Thank you for your valuable comments. We have revised our manuscript by adding “Experiments and theory have suggested versatile possible phases for adsorbed and confined water [30–48] which are closely to the aspects of various phenomena in materials science, geology, biology, tribology and nanotechnology. [10,49–51]” on line 27 of page 1.

Point 2. In Figure 6, the authors discuss an interesting relationship between the water dipole moment with the change in the structure of the surface where the water is attached. It will be helpful if the author can address whether the intermolecular distance between surface atoms (i.e. the solid density) also affects the dipole moment.

Response2:

Figure R1. Effect of the lattice constant on the parallel component of the effective dielectric constant of interfacial water at the fcc (100) surface for f = 1.0, respectively.

Thank you for your valuable comments. In order to study effect of the intermolecular distance between surface atoms or the solid density (l) on the interfacial parallel dielectric constant, we have carried on another two molecular dynamics simulations with a small uninform strain (±5%) for the fcc (100) surface at f = 1.0, respectively. The results are shown in Figure R1. For the fcc (100) surface, as l increases, first decreases from 115 at l = 0.372 nm to a minimum of 40 at l = 0.392 nm (the original lattice constant in the manuscript) and then increases to 84 at l = 0.412 nm. This is consistent with the results of our previous study [J. Phys. Chem. C, 2015, 119, 20409-20415], which has shown that the ordered water monolayer at interface is greatly dependent on the lattice constant. Therefore, for the fcc (100) surface, changing the lattice constant would affect the results.

Point 3. Not a critical issue, but I hope that the publisher, in particular the layout team, can help to typeset the "\varepsilon ||^{interfacial}" symbol because, in its current state, the symbol does not look well in Figure 6 caption (and on paragraph too). Alternatively, the authors can choose to simplify the symbol.

Looking forward to seeing the review published.

Response3: Thank you for your valuable comments. In the revised manuscript, we have changed the symbol of .

Reviewer 3 Report

As attached

Author Response

Chunlei Wang et al. reviewed on Ordered/Disordered Structures of Water at Solid/Liquid Interfaces. In this reason, ordered water structure is mainly attributed to the mismatching or matching between solid surface structures and water molecules surfaces. The physical so-called electric double layer (EDL) was usually induced by the charged surfaces when the aqueous electrolyte solution is present. However, the classical theory but neglecting the interfacial molecular water structures is still unable to fully describe the observed phenomenon under the assumptions, such as no ion-ion correlations and the homogeneous dielectric continuum of water. The present article having very interesting and important concepts of wetting behaviour which is very useful for design of novel energy storage system also and therefore I would recommend strongly for the publications in Crystals.

Thank you.

Response: Thank you for your valuable comments. We have also carefully revised our manuscript accordingly.

Round 2

Reviewer 1 Report

My comments have been addressed in a satisfactory way.